# CONSTRAINED PREFERENCE RLHF

## ABSTRACT

We study offline constrained reinforcement learning from human feedback with multiple preference oracles. Motivated by applications that trade off performance with safety or fairness, we aim to maximize target population utility subject to a minimum protected group welfare constraint. From pairwise comparisons collected under a reference policy, we estimate oracle-specific rewards via maximum likelihood and analyze how statistical uncertainty propagates through the dual program. We cast the constrained objective as a KL regularized Lagrangian whose primal optimizer is a Gibbs policy, reducing learning to a one-dimensional convex dual problem. We propose a dual-only algorithm that ensures high-probability constraint satisfaction and provide finite-sample performance guarantees for the resulting Gibbs policy. Our analysis shows how estimation error, data coverage, and constraint slack jointly affect feasibility and optimality.

## 1 INTRODUCTION

*Reinforcement Learning from Human Feedback* (RLHF) has rapidly become a cornerstone for aligning AI behavior with human preferences, especially when explicit reward specification is impractical or unreliable. Early studies in preference-based reinforcement learning demonstrated that effective policies can be derived directly from comparative human judgments (Jain et al., 2015; Busa-Fekete et al., 2014; Daniel et al., 2015; Wirth et al., 2016). Building on this foundation, deep-RL methods showed that reward models learned from pairwise feedback outperform hand-crafted rewards in complex domains such as games and robotics (Christiano et al., 2017). The same paradigm has proven successful for language modeling: preference-guided PPO (Ziegler et al., 2020), recursive reward modeling (Wu et al., 2021), and instruction-tuned systems like InstructGPT (Ouyang et al., 2022) substantially enhance the helpfulness, harmlessness, and truthfulness of large language models. These empirical successes have, in turn, spurred a growing body of theoretical work aimed at rigorously characterizing the statistical and computational properties of RLHF (Xiong et al., 2024; Ji et al., 2025; Kaufmann et al., 2024).

Theoretical work in RLHF began with (Novoseller et al., 2020), which minimized regret from trajectory comparisons using Dueling Posterior Sampling. Follow-up algorithms such as PPS and PEPS (Xu et al., 2020), PR-LSVI (Wu & Sun, 2024), and PARL (Chakraborty et al., 2024) established regret bounds and convergence guarantees. Other contributions include generalizing to rich dynamics and feedback (Chen et al., 2022), deriving minimax regret bounds (Saha et al., 2023), and analyzing policy optimization under linear and neural models (Du et al., 2024). Further developments expanded the feedback models (Zhu et al., 2023; Zhan et al., 2024), introduced interactive and feedback-efficient learning frameworks (Kong & Yang, 2022), and provided off-policy guarantees (Li et al., 2023), privacy protection (Chowdhury & Zhou, 2023), and algorithmic reductions from standard RL (Wang et al., 2023). Recent work sharpens safety and robustness via high-confidence constraint satisfaction (Chittepu et al., 2025) and convergence under unknown preference mappings (Zhang & Ying, 2025).

Most of the analysis so far focuses on a single Oracle model, where the reward is mainly dependent on the preferences given by a single oracle. Many practical deployments demand more than a single "helpfulness" objective: the system must maximize performance for its primary users while ensuring that the protected group achieves a predefined standard. This multi-objective alignment is essential for fairness, such as limiting cultural misappropriation or mitigating disparate impact across demographic groups (Siddique et al., 2023); for legal compliance, where decisions must adhere to open-textured regulations like those in the European Convention on Human Rights (Botskina,

2025); and for safety-critical operation, where behavior must remain within formally verified safety envelopes (Dai et al., 2024). These requirements can be formalized as a constrained RLHF problem involving two distinct reward oracles:

$$\max_{\pi} \mathbb{E}_{\pi}[r_1^{\star}] - \eta D_{\mathrm{KL}}(\pi \| \pi_0) \quad \text{s.t.} \quad \mathbb{E}_{\pi}[r_2^{\star}] \geq J_{\min},$$

where $r_1^{\star}$ reflects utility for the primary users, $r_2^{\star}$ quantifies the welfare of the protected group, $\pi_0$ is a reference policy, $J_{\min}$ denotes the minimum acceptable performance for the protected group, and $\eta > 0$ governs the trade-off between utility and deviation from the reference policy.

In this paper, we investigate *constrained RLHF* in the *offline* setting, where the learner receives a fixed batch of preference data consisting of paired comparisons annotated by both the primary and the protected populations. The learner must synthesize a policy without any further environment interaction. Our main contributions are:

- We present the first formal treatment of constrained RLHF with multiple reward oracles, integrating preference-based reward estimation and constraint satisfaction into a unified framework.
- We design a dual-only algorithm that jointly optimizes the policy and Lagrange multiplier, using reward estimates inferred from offline pairwise comparisons provided by both oracles.
- We provide non-asymptotic, sample-dependent and sample-independent guarantees showing how dataset coverage governs the optimality gap and constraint violation of the learned policy.

*To the best of our knowledge, this work is the first to establish finite-sample guarantees for constrained RLHF with multiple reward oracles in the offline setting.*

## 2 RELATED WORK

This section reviews two lines of prior work: we survey key results in RLHF, and we outline the main ideas in constrained RL. This context highlights the gap that our study addresses.

**RLHF:** A growing body of work has established theoretical guarantees for RLHF in both online and offline regimes. In the offline regime, Zhu et al. (2023) provided the first finite-sample guarantees for policies trained via maximum likelihood estimation under the Bradley–Terry model with linear rewards (Bradley & Terry, 1952). Building on this, Zhan et al. (2024) introduced the FREEHAND algorithm, which generalizes earlier approaches by allowing a broader class of reward functions and feedback models. Kong & Yang (2022) developed feedback-efficient frameworks that incorporate human-in-the-loop reward specification. Li et al. (2023) provided the first off-policy analysis for offline RLHF via the DCPPO algorithm. Zhu et al. (2024) addressed overfitting and overoptimization in reward learning and proposed the IDS algorithm for improved robustness. Chowdhury & Zhou (2023) studied privacy-preserving reward learning in preference-based RL. Wang et al. (2023) presented reduction-based approaches that adapt reward-based RL algorithms to the RLHF setting, showing how theoretical guarantees transfer. Finally, Zhou et al. (2025) unified the analysis of privacy and robustness in offline RLHF.

**Constraint RL:** Constrained RL traces dates back to CMDP formulations of the 1970s–1990s (Kolesar, 1970; Ross, 1989; Altman, 1999). Foundational optimization ideas such as Lagrangian relaxation (Everett, 1963; Shapiro, 1979) and primal-dual updates (Altman, 1998; Efroni et al., 2020; Bertsekas, 2016) motivated algorithms that embed multipliers directly into RL procedures (Zheng & Ratliff, 2020; Ding et al., 2020; Ying et al., 2022). In the offline setting, guarantees are now available for primal–dual critic methods (Hong et al., 2024), LP-based algorithms under partial coverage (Hong & Tewari, 2025), and multi-constraint primal policy optimization (Guan et al., 2024).

## 3 FORMULATION

In this section, we present the problem formulation. We extend the standard preference-based RLHF setup (Ouyang et al., 2022; Xiong et al., 2024) to the constrained setting. Let $\mathcal{X}$ be the finite prompt space and $\mathcal{A}$ the finite response space. A prompt $x \sim d_0$ is first sampled from a fixed distribution $d_0$. Conditional on $x$, two independent responses $a, a' \in \mathcal{A}$ from the reference policy $\pi_0$ are

produced. Feedback is then collected from two independent human-preference oracles: a target-population oracle $o_1$ and a protected-group oracle $o_2$. Each oracle's binary preference, following the Bradley–Terry model, is modeled as a Bernoulli random variable whose success probability is given by a logistic function of the latent reward gap:

$$o_k(x, a, a') \sim \text{Ber}\left(\sigma(r_k^*(x, a) - r_k^*(x, a'))\right), \quad k \in \{1, 2\},$$

where $r_k^* : \mathcal{X} \times \mathcal{A} \to \mathbb{R}$ is the unknown reward function for oracle $k$ and $\sigma(z) = (1 + e^{-z})^{-1}$ is the logistic link. Specifically, $o_k(x, a, a') = 1$ indicates that oracle $o_k$ prefers action $a$ over action $a'$, denoted as $a \succ_k a'$.

We operate in the offline setting, where the learner has access to a dataset $\mathcal{D}_N = \{(x_i, a_i^{(1)}, a_i^{(2)}, y_{i,1}, y_{i,2})\}_{i=1}^N$ consisting of prompts, response pairs, and corresponding binary preferences. Each tuple $(x_i, a_i^{(1)}, a_i^{(2)})$ is drawn *i.i.d.* from the distribution $(x, a, a') \sim \mu_0 := d_0(x)\pi_0(a|x)\pi_0(a'|x)$. For each triplet, the preferences $y_{i,1}, y_{i,2} \in \{0, 1\}$ are independently sampled from the two oracles $o_1$ and $o_2$. More specifically,

$$\mathbb{P}(y_{i,k} = 1 | x, a_i^{(1)}, a_i^{(2)}) = \mathbb{P}\left(a_i^{(1)} \succ_k a_i^{(2)} \big| x, a_i^{(1)}, a_i^{(2)}\right) = \sigma(r_k^*(x_i, a_i^{(1)}) - r_k^*(x_i, a_i^{(2)}))$$

The learner seeks a policy $\pi^* : \mathcal{X} \to \Delta(\mathcal{A})$ that maximizes the expected reward of the target population, remains sufficiently close to a reference policy $\pi_0$, and ensures a minimum level of welfare for the protected group. Here, $\Delta(\mathcal{A})$ denotes the set of all probability distributions over the response set $\mathcal{A}$. Formally, the goal is to solve the following constrained optimization problem:

$$\max_{\pi \in \Pi} \ \mathbb{E}_{x \sim d_0} \left[ \mathbb{E}_{a \sim \pi(\cdot|x)} \left[ r_1^*(x, a) \right] - \eta D_{\text{KL}} \left( \pi(\cdot|x) \| \pi_0(\cdot|x) \right) \right]$$

$$\text{s.t.} \ \mathbb{E}_{x \sim d_0} \left[ \mathbb{E}_{a \sim \pi(\cdot|x)} \left[ r_2^*(x, a) \right] \right] \geq J_{\min},$$

where $\Pi$ denotes the set of all policies $\pi : \mathcal{X} \to \Delta(\mathcal{A})$, $J_{\min}$ denotes the minimum acceptable reward for the protected group, and $\eta > 0$ controls the trade-off between utility and divergence from the reference policy. For notational convenience, we rewrite the problem in abstract form:

$$\max_{\pi \in \Pi} \ J(\pi) \quad \text{s.t.} \quad c(\pi) \leq 0, \tag{1}$$

where $J(\pi)$ denotes the regularized target reward objective, and $c(\pi) := J_{\min} - \mathbb{E}_{x \sim d_0} \left[ \mathbb{E}_{a \sim \pi(\cdot|x)} [r_2^*(x, a)] \right]$ is the constraint function. Throughout the paper, we adopt the short-hand notation $\mathbb{E}_\pi := \mathbb{E}_{x \sim d_0} \mathbb{E}_{a \sim \pi(\cdot|x)}$ and $\mathbb{E} := \mathbb{E}_{x \sim d_0}$, unless stated otherwise.

Following Xiong et al. (2024); Zhu et al. (2023), we impose the following assumptions on the reference policy and the reward functions.

**Assumption 1** (Full coverage). *For every $x \in \mathcal{X}$, the reference policy $\pi_0(\cdot \mid x)$ has full support over the finite action space $\mathcal{A}$.*

**Assumption 2** (Linear reward). *For each $k \in \{1, 2\}$, the latent reward is assumed to be linear in a known feature map $\phi$, i.e., $r_k^*(x, a) = \langle \theta_k^*, \phi(x, a) \rangle$, with $\|\phi(x, a)\| \leq 1$ for all $(x, a)$ and $\|\theta_k^*\| \leq B$.*

**Assumption 3** (Identifiability). *Let $\Delta(x; a, a') := \phi(x, a) - \phi(x, a')$ and define the population difference covariance matrix $\Sigma_\infty := \mathbb{E}_{(x, a, a') \sim \mu_0}[\Delta(x; a, a')\Delta(x; a, a')^\top]$. We assume that $\Sigma_\infty \succ 0$, or equivalently, $\text{span}\{\Delta(x; a, a') : x \in \mathcal{X}, a, a' \in \mathcal{A}\} = \mathbb{R}^d$, so that $\theta_k^*$ for $k \in \{1, 2\}$ is uniquely identifiable from pairwise comparisons.*

Notice that the identifiability assumption is necessary for the uniqueness of $\theta_k$. Without it, there exists a nonzero vector $v$ such that $v^\top \Delta(x; a, a') = 0$, for all $x \in \mathcal{X}$ and $a, a' \in \mathcal{A}$. In that case, the likelihood is invariant along the ray $\theta_k + tv$ for $t \in \mathbb{R}$, so the solution set for $\theta_k$ is an affine line rather than a point.

## 4 ANALYSIS

In this section, we analyze the constrained optimization problem introduced above. First, using the offline preference dataset $\mathcal{D}_N$, we obtain maximum-likelihood estimators $\widehat{r}_1$ and $\widehat{r}_2$ of the latent reward functions $r_1^*$ and $r_2^*$. We then construct the corresponding Lagrangian, derive the associated dual problem, and verify the convexity conditions required for strong duality.

## 4.1 REWARD ESTIMATION

Under Assumption 2 and 3 and the conditional independence of oracle preferences given $(x, a_i^{(1)}, a_i^{(2)})$, the joint log-likelihood of the dataset $\mathcal{D}_N$ is

$$\ell_{\mathcal{D}_N}(\theta_1, \theta_2) = \sum_{k \in \{1,2\}} \sum_{i=1}^N \ell_i^{(k)}(\theta_k),$$

where $\ell_i^{(k)}(\theta_k) = y_{i,k} \log \sigma(\langle \theta_k, \Delta_i \rangle) + (1 - y_{i,k}) \log \sigma(-\langle \theta_k, \Delta_i \rangle)$, and $\Delta_i := \phi(x_i, a_i^{(1)}) - \phi(x_i, a_i^{(2)})$. Maximizing the individual log-likelihoods $\sum_{i=1}^N \ell_i^{(k)}(\theta_k)$ for each $k \in \{1, 2\}$ yields the maximum-likelihood estimators $\widehat{\theta}_k$. By Lemma 3.1 of Zhu et al. (2023), the in-sample estimation error satisfies, with probability at least $1 - 2\delta$ for every $k \in \{1, 2\}$,

$$\|\widehat{\theta}_k - \theta_k^*\|_{\Sigma_{N,\mathrm{reg}}} \leq C \sqrt{\frac{d + \log(1/\delta)}{\gamma^2 N} + \lambda_{\mathrm{reg}} B^2} =: \beta_N$$

where $\| \cdot \|_{\Sigma_{N,\mathrm{reg}}}$ denotes the Mahalanobis norm induced by the regularized empirical covariance matrix $\Sigma_{N,\mathrm{reg}} := \Sigma_{\mathcal{D}_N} + \lambda_{\mathrm{reg}} I$ with

$$\Sigma_{\mathcal{D}_N} = \frac{1}{N} \sum_{i=1}^N \Delta_i \Delta_i^\top.$$

Here $d$ is the feature dimension, $B$ is an upper bound on the norm of the reward parameters, $\lambda_{\mathrm{reg}} > 0$ is the regularization parameter, $\gamma = 1/(2 + e^{-B} + e^B)$ reflects the curvature of the logistic likelihood, and $C > 0$ is a problem dependent constant.

We factor the likelihood under conditional independence for clarity, but our guarantees are agnostic to this assumption: the stated rates hold under arbitrary dependence and thus represent the worst case over correlation structures between $\theta_1^*$ and $\theta_2^*$. When prior correlation or shared latent structure is known, joint estimation can exploit it to tighten constants and reduce sample complexity. Absent such knowledge, the independence-based presentation serves as a baseline that avoids modeling or validating dependence.

## 4.2 DUAL PROBLEM ANALYSIS

Let $\mathcal{L}(\pi, \lambda)$ be the Lagrangian of the constrained problem (1). We have

$$\mathcal{L}(\pi, \lambda) = \mathbb{E}_\pi[r_1^*(x, a) + \lambda r_2^*(x, a)] - \eta \mathbb{E}[D_{\mathrm{KL}}(\pi(\cdot|x)\|\pi_0(\cdot|x))] - \lambda J_{\min}$$

The associated dual problem is therefore $\min_{\lambda \geq 0} \max_\pi \mathcal{L}(\pi, \lambda)$.

Observe that $\Pi$ is a convex set. Moreover, for every $x \in \mathcal{X}$, the KL-divergence $D_{\mathrm{KL}}(\pi(\cdot \mid x)\|\pi_0(\cdot \mid x))$ is strictly convex in $\pi(\cdot \mid x)$. Hence, the primal problem is a strictly concave maximization with an affine constraint, and under Slater's condition, the strong duality holds.

**Assumption 4** (Slater's condition). *There exists a policy $\tilde{\pi} \in \Pi$ and a slack $\rho > 0$ such that*

$$\mathbb{E}_{\tilde{\pi}}\left[r_2^*(x, a)\right] \geq J_{\min} + \rho.$$

Slater's condition can be verified using the greedy policy for $r_2^*$. With the estimate $\widehat{\theta}_2$, the slack $\rho$ can be approximated with high probability, since with probability at least $1 - \delta$ we have $\|\widehat{\theta}_2 - \theta_2^*\|_{\Sigma_{\mathcal{D}_N,\mathrm{reg}}} \leq \beta_N$. Thus, if $\beta_N$ is sufficiently small, the slack can be estimated using $\mathbb{E}_{\tilde{\pi}}[\widehat{r}_2(x, a)] - J_{\min}$, where $\tilde{\pi}$ is the greedy policy for $\widehat{r}_2(x, a) = \langle \widehat{\theta}_2, \phi(x, a) \rangle$.

**Corollary 1.** *Let $\tilde{\pi}$ be the greedy policy w.r.t. $\widehat{\theta}_2$, i.e., $\tilde{\pi}(a|x) = \mathbf{1}\{a \in \arg\max_{a'}\langle \widehat{\theta}_2, \phi(x, a') \rangle\}$. With probability at least $1 - \delta$,*

$$\mathbb{E}_{\tilde{\pi}}[r_2^*(x, a)] \geq \mathbb{E}_{\tilde{\pi}}[\widehat{r}_2(x, a)] - \frac{\beta_N}{\sqrt{\lambda_{\min}(\Sigma_{N,\mathrm{reg}})}}.$$

*Hence, if the right-hand side is strictly larger than $J_{\min}$, Assumption 4 holds with slack*

$$\widehat{\rho} = \tfrac{1}{2}\left(\mathbb{E}_{\tilde{\pi}}[\widehat{r}_2(x, a)] - \frac{\beta_N}{\sqrt{\lambda_{\min}(\Sigma_{N,\mathrm{reg}})}} - J_{\min}\right).$$

*Proof.* Applying Cauchy–Schwarz, the confidence bound $\|\theta_2^* - \widehat{\theta}_2\|_{\Sigma_{N,\text{reg}}} \leq \beta_N$, and $\|\phi(x, a)\| \leq 1$ yields the inequality. $\qquad\square$

This estimate can be used to bound the optimal dual parameter. Since $J(\pi)$ is strictly concave and the feasible set is convex, the constrained problem admits a unique optimal policy $\pi^\star \in \Pi$.

Consider the dual function $g(\lambda) = \max_\pi \mathcal{L}(\pi, \lambda)$. By standard results for KL-regularized objectives (e.g., Zhang (2023)), the maximizer $\pi_\lambda^* = \arg\max_\pi \mathcal{L}(\pi, \lambda)$ has the Gibbs (Boltzmann) form

$$\pi_\lambda^*(a|x) = \frac{\pi_0(a|x) \exp\left(\frac{1}{\eta} \langle \theta_1^* + \lambda\theta_2^*, \phi(x, a) \rangle\right)}{Z_\lambda(x)}$$

where $Z_\lambda(x)$ is the normalizing constant, also referred to as the partition function. Having a closed-form solution to the dual problem enables an efficient dual-only algorithm. Following Xiong et al. (2024), we assume access to a "Policy Improvement Oracle".

**Definition 1** (Policy Improvement Oracle (Xiong et al., 2024))**.** *For reward function $r : \mathcal{X} \times \mathcal{A} \to \mathbb{R}$ and a reference policy $\pi_0$, for all $x \in \mathcal{X}$, we can compute the Gibbs policy:*

$$\pi_r(\cdot \mid x) := \arg\max_{\pi \in \Pi} \mathbb{E}_{a \sim \pi(\cdot|x)} \left[ r(x, a) - \eta \log \frac{\pi(a \mid x)}{\pi_0(a \mid x)} \right] \quad \propto \pi_0(\cdot \mid x) \cdot \exp\left( \frac{1}{\eta} r(x, \cdot) \right).$$

Next, we analyze the properties of the dual function $g(\cdot)$. By the envelope theorem, we have

$$g'(\lambda) = \mathbb{E}_{\pi_\lambda^*}[r_2^*(x, a)] - J_{\min}.$$

Since $\pi_\lambda^*$ belongs to an exponential family, its mean parameter is Lipschitz continuous given boundedness of its sufficient statistics. Consequently, the derivative of the dual function $g'(\lambda)$ is Lipschitz continuous (Wainwright & Jordan, 2007; Brown, 1986).

**Lemma 1.** *The derivative $g'(\lambda)$ is Lipschitz continuous with Lipschitz constant $L = \frac{B^2}{\eta}$.*

*Proof.* Fix $x \in \mathcal{X}$, and let $A(\lambda) = \log Z_\lambda(x)$ denote the log-partition function of $\pi_\lambda^*(\cdot \mid x)$. Then

$$\frac{d}{d\lambda} A(\lambda) = \eta^{-1} \mathbb{E}_{\pi_\lambda^*(\cdot|x)}[r_2^*(x, a)], \qquad \frac{d^2}{d\lambda^2} A(\lambda) = \eta^{-2} \text{Cov}_{\pi_\lambda^*(\cdot|x)}(r_2^*(x, a)).$$

By Assumption 2, the reward function is bounded by $B$, so the covariance is bounded by $B^2$, and hence the derivative of $A$ is Lipschitz with constant at most $\frac{B^2}{\eta^2}$. $\qquad\square$

The same properties also hold for the empirical dual function $\widehat{g}(\lambda) = \max_\pi \mathcal{L}(\pi, \lambda; \widehat{\theta}_1, \widehat{\theta}_2)$, where $\mathcal{L}(\pi, \lambda; \widehat{\theta}_1, \widehat{\theta}_2)$ denotes the Lagrangian of the primal problem (1) with the true parameters $\theta_1^*$ and $\theta_2^*$ replaced by their statistical estimates. In this case, $\pi_\lambda^*$ is replaced in the proof by $\widehat{\pi}_\lambda$, the policy that attains the maximum in $\widehat{g}(\lambda)$. Next, we quantify the gap between $g(\lambda)$ and $\widehat{g}(\lambda)$, as well as their derivatives, in terms of the estimation errors of $\theta_1^*$, $\theta_2^*$, and the regularized sample covariance matrix. These bounds make explicit how statistical uncertainty propagates through the dual program.

**Lemma 2.** *For any $\lambda \geq 0$ we have with probability at least $1 - 2\delta$, we have*

$$|\widehat{g}(\lambda) - g(\lambda)| \leq \frac{(1 + \lambda)\beta_N}{\sqrt{\lambda_{\min}(\Sigma_{N,\text{reg}})}}, \quad \text{and} \quad |\widehat{g}'(\lambda) - g'(\lambda)| \leq \frac{\beta_N}{\sqrt{\lambda_{\min}(\Sigma_{N,\text{reg}})}} \left( 1 + \frac{B(1 + \lambda)}{\eta} \right),$$

*where $\lambda_{\min}(\Sigma_{N,\text{reg}}) > 0$ is the smallest eigenvalue of regularized sample covariance matrix.*

*proof.* Notice that $|g(\lambda) - \widehat{g}(\lambda)| \leq \max_{\pi \in \{\pi_\lambda^*, \widehat{\pi}_\lambda\}} |\mathcal{L}(\pi, \lambda; \theta_1^*, \theta_2^*) - \mathcal{L}(\pi, \lambda; \widehat{\theta}_1, \widehat{\theta}_2)|$. This implies the bound

$$\max_{\pi \in \{\pi_\lambda^*, \widehat{\pi}_\lambda\}} \mathbb{E}_\pi \left[ \|\phi(x, a)\|_{\Sigma_{N,\text{reg}}^{-1}} \right] \cdot \|\theta_1^* - \widehat{\theta}_1 + \lambda(\theta_2^* - \widehat{\theta}_2)\|_{\Sigma_{N,\text{reg}}}.$$

Finally, by the MLE error and the bound $\|\phi(x,a)\|_{\Sigma_{N,\text{reg}}^{-1}} \leq \|\phi(x,a)\|/\sqrt{\lambda_{\min}(\Sigma_{N,\text{reg}})}$, the result follows. To bound the difference of derivatives, notice that

$$|\widehat{g}'(\lambda) - g'(\lambda)| = |\mathbb{E}_{\widehat{\pi}_\lambda}[\widehat{r}_2] - \mathbb{E}_{\pi_\lambda^*}[r_2^*]| \leq |\mathbb{E}_{\widehat{\pi}_\lambda}[\widehat{r}_2 - r_2^*]| + |\mathbb{E}_{\widehat{\pi}_\lambda}[r_2^*] - \mathbb{E}_{\pi_\lambda^*}[r_2^*]|.$$

The first term can be bounded as before, while the second is bounded as follows:

$$|\mathbb{E}_{\widehat{\pi}_\lambda}[r_2^*] - \mathbb{E}_{\pi_\lambda^*}[r_2^*]| \leq \|\theta_2^*\| \cdot \|\mathbb{E}_{a\sim\widehat{\pi}_\lambda(\cdot|x)}[\phi(x,a)] - \mathbb{E}_{a\sim\pi_\lambda^*(\cdot|x)}[\phi(x,a)]\|$$

$$\overset{(a)}{\leq} B\|\frac{1}{\eta}(\widehat{\theta}_1 + \lambda\widehat{\theta}_2) - \frac{1}{\eta}(\theta_1^* + \lambda\theta_2^*)\| \leq \frac{(1+\lambda)\beta_N}{\eta\sqrt{\lambda_{\min}(\Sigma_{N,\text{reg}})}}.$$

where (a) follows from the boundedness of $\theta_2^*$ and an argument similar to Lemma 1. $\qquad\square$

The bounds above yield practical, data–dependent upper bound on the accuracy of estimating $g(\lambda)$ and its derivative. To decouple these guarantees from a particular sample, note that $\Sigma_N \xrightarrow{\text{a.s.}} \Sigma_\infty$ by the law of large numbers. Moreover, standard covariance concentration for bounded feature differences implies that $\|\Sigma_N - \Sigma_\infty\|_{op}$ is small with high probability, where $\|\cdot\|_{op}$ denotes the operator norm. Using this, the following proposition provides a high–probability, sample–independent change–of–norm relations.

**Lemma 3.** *With probability at least $1 - \delta$, the following bounds hold for any $v \in \mathbb{R}^d$:*

$$\frac{\|v\|_{\Sigma_{N,\text{reg}}}}{\zeta_{\max}(\delta,N)} \leq \|v\|_2 \leq \frac{\|v\|_{\Sigma_{N,\text{reg}}}}{\zeta_{\min}(\delta,N)},$$

*and similarly*

$$\zeta_{\min}(\delta,N) \cdot \|v\|_{\Sigma_{N,\text{reg}}^{-1}} \leq \|v\|_2 \leq \zeta_{\max}(\delta,N) \cdot \|v\|_{\Sigma_{N,\text{reg}}^{-1}}.$$

*Here $\zeta_{\min}(\delta,N)$ and $\zeta_{\max}(\delta,N)$ quantify the deviation of the smallest and largest eigenvalues of $\Sigma_{N,\text{reg}}$ from their asymptotic counterparts, respectively, and are given by*

$$\zeta_{\max}(\delta,N) := \sqrt{(1 + \overline{\varepsilon}_N(\delta))\lambda_{\max}(\Sigma_{\mathcal{D}_\infty}) + \lambda_{reg}},$$

$$\zeta_{\min}(\delta,N) := \sqrt{(1 - \underline{\varepsilon}_N(\delta))\lambda_{\min}(\Sigma_{\mathcal{D}_\infty}) + \lambda_{reg}}.$$

*The error terms are*

$$\overline{\varepsilon}_N(\delta) := CK^2\left(\sqrt{\frac{d + \log(\frac{2}{\delta})}{N}} + \frac{d + \log(\frac{2}{\delta})}{N}\right) \quad and \quad \underline{\varepsilon}_N(\delta) := \frac{\lambda_{\max}(\Sigma_{\mathcal{D}_\infty})}{\lambda_{\min}(\Sigma_{\mathcal{D}_\infty})}\overline{\varepsilon}_N(\delta).$$

*Proof.* By Assumption 2, we have $\|\phi(x,a)\|_2 \leq 1$. Therefore $\|\Delta\|_2 \leq 2$. Hence $\Delta$ is a sub-Gaussian vector with parameter $K = O(1)$. Then by Theorem 4.7.1 and Remark 4.7.3 in Vershynin (2018) we have with probability $1 - \delta$, we have $\|\Sigma_{\mathcal{D}_N} - \Sigma_{\mathcal{D}_\infty}\|_{op} \leq \overline{\varepsilon}_N(\delta)\|\Sigma_{\mathcal{D}_\infty}\|_{op}$.

Since $\Sigma_{\mathcal{D}_\infty}$ and $\Sigma_{\mathcal{D}_N}$ are both positive semi-definite, by triangle inequality we have

$$\lambda_{\max}(\Sigma_{\mathcal{D}_N}) \leq \lambda_{\max}(\Sigma_{\mathcal{D}_\infty}) + \lambda_{\max}(\Sigma_{\mathcal{D}_N} - \Sigma_{\mathcal{D}_\infty}) \leq (1 + \overline{\varepsilon}_N(\delta))\lambda_{\max}(\Sigma_{\mathcal{D}_\infty})$$

Furthermore, by a corollary (spectral stability) of Weyl's inequality, we have

$$|\lambda_{\min}(\Sigma_{\mathcal{D}_N}) - \lambda_{\min}(\Sigma_{\mathcal{D}_\infty})| \leq \|\Sigma_{\mathcal{D}_N} - \Sigma_{\mathcal{D}_\infty}\|_{op} \leq \underline{\varepsilon}_N(\delta)\lambda_{\min}(\Sigma_{\mathcal{D}_\infty})$$

Therefore $\lambda_{\min}(\Sigma_{\mathcal{D}_N}) \geq (1 - \underline{\varepsilon}_N)\lambda_{\min}(\Sigma_{\mathcal{D}_\infty})$. Combining the inequalities and noting $\lambda_i(\Sigma_{N,\text{reg}}) = \lambda_i(\Sigma_{\mathcal{D}_N}) + \lambda_{\text{reg}}$, the result follows from the definition of the Mahalanobis norm. $\quad\square$

Combining Lemma 3 with Lemma 2 and using the norm equivalences to replace sample–dependent spectral terms by population–level quantities yields the following: with probability at least $1 - 3\delta$ we have

$$|\widehat{g}(\lambda) - g(\lambda)| \leq \frac{(1+\lambda)\beta_N}{\zeta_{\min}(\delta,N)}, \quad \text{and} \quad |\widehat{g}'(\lambda) - g'(\lambda)| \leq \frac{\beta_N}{\zeta_{\min}(\delta,N)}\left(1 + \frac{B(1+\lambda)}{\eta}\right).$$

For brevity, we define the value and derivative error envelopes $\mathcal{E}_g(\lambda)$ and $\mathcal{E}_{g'}(\lambda)$. Depending on the use case, these envelopes may represent either the data-dependent or the data-independent versions:

$$\mathcal{E}_g(\lambda) := \frac{(1+\lambda)\beta_N}{\left\{ \zeta_{\min}(\delta, N) \text{ or } \sqrt{\lambda_{\min}(\Sigma_{N,\text{reg}})} \right\}}, \qquad \mathcal{E}_{g'}(\lambda) := \frac{\left(1 + B(1+\lambda)\eta^{-1}\right)\beta_N}{\left\{ \zeta_{\min}(\delta, N) \text{ or } \sqrt{\lambda_{\min}(\Sigma_{N,\text{reg}})} \right\}}.$$

Finally, since $g'$ is $L$–Lipschitz with $L = B^2/\eta$, these envelopes can be extended uniformly over $[0, \Lambda]$ via a standard $\varepsilon$–net argument.

Next, we establish convexity properties of $g(\lambda)$; the same conclusions apply to $\widehat{g}(\lambda)$ with the parameter $m_g(\cdot)$ replaced by $m_{\widehat{g}}(\cdot)$ defined analogously. As in Lemma 1, one can further bound the difference between $m_{\widehat{g}}(\cdot)$ and $m_g(\cdot)$.

**Proposition 1.** *Under the standing assumptions, the dual function $g$ is $m_g(\Lambda)$–strongly convex on $[0, \Lambda]$ where*

$$m_g(\Lambda) := \frac{1}{\eta} \inf_{\lambda \in [0,\Lambda]} \mathbb{E}\left[ \operatorname{Var}_{a \sim \pi^*_\lambda(\cdot|x)}\left(r^*_2(x,a)\right) \right] > 0.$$

*Proof.* Analogous to the proof of Lemma 1, we have $g''(\lambda) = \frac{1}{\eta}\mathbb{E}\left[ \operatorname{Var}_{a \sim \pi^*_\lambda(\cdot|x)}\left(r^*_2(x,a)\right) \right]$. Since $\eta > 0$ and $\pi_0(\cdot|x)$ has full support on $\mathcal{A}$, the Gibbs policy $\pi^*_\lambda(\cdot|x)$ also has full support for every $x$ and every finite $\lambda$. Hence, for any $x$ where $r^*_2(x,\cdot)$ is non-constant, $\operatorname{Var}_{\pi^*_\lambda(\cdot|x)}(r^*_2(x,a)) > 0$; by the positive–measure assumption, the expectation over $x$ is strictly positive for every $\lambda \in [0, \Lambda]$. The map $\lambda \mapsto g''(\lambda)$ is continuous, so on the compact interval $[0, \Lambda]$, $m_g(\Lambda) = \inf_{\lambda \in [0,\Lambda]} g''(\lambda)$ is attained and, since $g''(\lambda) > 0$ for all $\lambda$, we have $m_g(\Lambda) > 0$. Therefore, $g$ is $m_g(\Lambda)$–strongly convex on $[0, \Lambda]$. $\square$

With these properties in place, we can now present the main result of this section.

**Theorem 1.** *Under the standing assumptions, let $\pi^*$ denote the optimal primal policy that solves the primal problem* (1)*, and let $\lambda^* := \arg\min_{\lambda \geq 0} g(\lambda)$. Then $\pi^* = \pi^*_{\lambda^*}$. Moreover, $\lambda^*$ admits the following upper bounds:*

**Deterministic bound.** *Let $B$ be as in Assumption 2, and let $\tilde{\pi}$ and $\rho$ be as in Assumption 4. Define $\Lambda = \frac{B - J(\tilde{\pi})}{\rho}$. We have*

$$\lambda^* \leq \min\left\{ \Lambda, \frac{[-g'(0)]_+}{m_g(\Lambda)} \right\}.$$

**Data–driven bound.** *Let $B$ be as in Assumption 2, and let $\tilde{\pi}$ and $\rho$ be as in Corollary 4. Define $\Lambda = \rho^{-1}\left( B + \frac{\beta_N}{\sqrt{\lambda_{\min}(\Sigma_{N,\text{reg}})}} - \widehat{J}(\tilde{\pi}) \right)$. Then, with probability at least $1 - 3\delta$,*

$$\lambda^* \leq \min\left\{ \Lambda, \frac{[-\widehat{g}'(0) + \mathcal{E}_{g'}(0)]_+}{m_g(\Lambda)} \right\},$$

*where $\widehat{F}(\cdot)$ is defined analogously to $F(\cdot)$ but with $\theta^*_1$ and $\theta^*_2$ replaced by their estimates.*

*Proof.* By Slater's condition, strong duality holds and there exists $\lambda^* \geq 0$ such that $J(\pi^*) = g(\lambda^*)$, and $(\pi^*, \lambda^*)$ is a saddle point: $\mathcal{L}(\pi^*, \lambda) \geq \mathcal{L}(\pi^*, \lambda^*) \geq \mathcal{L}(\pi, \lambda^*)$ for all $\pi \in \Pi$ and $\lambda \geq 0$. The right inequality implies $\pi^* \in \arg\max_\pi \mathcal{L}(\pi, \lambda^*)$, and by uniqueness we have $\pi^* = \pi^*_{\lambda^*}$.

**Deterministic bound.** By strong duality, $B \geq g(\lambda^*) \geq \mathcal{L}(\tilde{\pi}, \lambda^*) \geq J(\tilde{\pi}) + \lambda^*\rho$. Thus $\lambda^* \leq \frac{B - J(\tilde{\pi})}{\rho} = \Lambda$. On the other hand, by strong convexity of $g(\cdot)$ we have, for all $\lambda \geq 0$, $g'(\lambda) \geq g'(0) + m_g(\Lambda)\lambda$. Substituting $\lambda^*$ and using $g'(\lambda^*) = 0$ yields the desired bound.

**Data–driven bound.** The result follows from Corollary 1, together with the facts that, with probability at least $1 - \delta$, $J(\tilde{\pi}) \geq \widehat{J}(\tilde{\pi}) - \frac{\beta_N}{\sqrt{\lambda_{\min}(\Sigma_{N,\text{reg}})}}$, and that, with probability at least $1 - 2\delta$, $g'(0) \leq \widehat{g}'(0) - \mathcal{E}_{g'}(0)$. Here, similar to Lemma 1, one can replace $m_g(\Lambda)$ with $m_{\widehat{g}}(\Lambda)$ at the cost of introducing an additional error term. $\square$

---

**Algorithm 1** Projected Gradient Descent (Dual)

---

**Require:** MLEs $\widehat{\theta}_1, \widehat{\theta}_2$ from $\mathcal{D}$; step size $\alpha$; constraint level $J_{\min}$; projection radius $R$; iterations $T$
**Ensure:** Approximate dual minimizer $\bar{\lambda}_T$
 1: Initialize $\lambda_0 \leftarrow 0$
 2: **for** $t = 0$ to $T - 1$ **do**
 3:    Policy Optimization: $\widehat{\pi}_{\lambda_t}(a|x) \propto \pi_0(a|x) \exp\left(\frac{1}{\eta}\langle\widehat{\theta}_1 + \lambda_t\widehat{\theta}_2, \phi(x,a)\rangle\right)$.
 4:    Gradient Estimation: $\widehat{g}'(\lambda_t) \leftarrow \mathbb{E}_{x\sim d_0}\mathbb{E}_{a\sim\widehat{\pi}_{\lambda_t}(\cdot|x)}\left[\langle\widehat{\theta}_2, \phi(x,a)\rangle\right] - J_{\min}$.
 5:    Projected Gradient Descent: $\lambda_{t+1} \leftarrow \mathrm{Proj}_{[0,R]}\left(\lambda_t - \alpha\widehat{g}'(\lambda_t)\right)$.
 6: **end for**
 7: **return** $\bar{\lambda}_T \leftarrow \frac{1}{T}\sum_{i=0}^{T-1}\lambda_t$

---

## 5 Algorithm

We now present our dual-only algorithm for solving the constrained RLHF problem. Our approach exploits the closed-form solution of the KL-regularized objective to reduce the constrained optimization to a one-dimensional convex problem over the dual variable. The unique Gibbs form of the optimal policy eliminates the need for complex primal-dual iterations. Instead, we minimize the dual via projected gradient descent on a high-probability domain $[0, R]$ for $\lambda^*$. With MLEs $\widehat{\theta}_1, \widehat{\theta}_2$ and a step size $\alpha = 1/L = \eta/B^2$ (from the Lipschitz constant of $g'$), the algorithm outputs an approximately optimal dual parameter that induces the corresponding Gibbs policy. Each iteration of the algorithm performs three steps: (i) form the current Gibbs policy, (ii) estimate the dual gradient, and (iii) take a projected gradient step, ensuring $\lambda$ remains in the range where our guarantees apply.

**Theorem 2.** *Under the standing assumptions, Algorithm 1, with projection radius $R$ chosen as in Theorem 1 and step size $\alpha = \frac{\eta}{B^2}$, yields:*

$$g(\bar{\lambda}_T) - g(\lambda^*) \leq 2\mathcal{E}_g(R) + \frac{B^2R^2}{2\eta T}, \qquad (J_{\min} - \mathbb{E}_{\pi^*_{\bar{\lambda}_T}}[r_2^*(x,a)])_+ \leq \mathcal{E}_{g'}(R) + \frac{B^2R}{\eta\sqrt{T}},$$

$$J(\pi^*) - J(\pi^*_{\bar{\lambda}_T}) \leq 2\mathcal{E}_g(R) + \frac{B^2R^2}{2\eta T} + R\left(\mathcal{E}_{g'}(R) + \frac{B^2R}{\eta\sqrt{T}}\right),$$

*with probability inherited from choice of $R$.*

*Proof.* The proof follows from standard projected gradient descent analysis combined with our concentration results. For the dual sub-optimality we decompose into two terms. The first term is upper bounded by our concentration $g(\bar{\lambda}_T) - \widehat{g}(\bar{\lambda}_T) + \widehat{g}(\lambda^*) - g(\lambda^*) \leq 2\mathcal{E}_g(R)$ and the second by standard results in projected gradient descent $\widehat{g}(\bar{\lambda}_T) - \widehat{g}(\lambda^*) \leq \widehat{g}(\bar{\lambda}_T) - \widehat{g}(\widehat{\lambda}^*) \leq \frac{B^2R^2}{2\eta T}$ where we use $\widehat{g}(\widehat{\lambda}^*) \leq \widehat{g}(\lambda^*)$ and Lipschitz parameter $L = \frac{B^2}{\eta}$ with step size $\frac{1}{L}$. For constraint violation, we decompose $|g'(\bar{\lambda}_T)|$ into two pieces. The first bounded by our concentration $|g'(\bar{\lambda}_T) - \widehat{g}'(\bar{\lambda}_T)| \leq \mathcal{E}_{g'}(R)$ and the second again by standard results in projected gradient descent $|\widehat{g}'(\bar{\lambda}_T)| \leq \sqrt{2L(\widehat{g}(\bar{\lambda}_T) - \widehat{g}(\widehat{\lambda}^*))} \leq \frac{B^2R}{\eta\sqrt{T}}$. For primal sub-optimality we note that $g(\lambda) = \mathcal{L}(\pi^*_\lambda, \lambda)$ and $J(\pi^*_{\bar{\lambda}_T}) = g(\bar{\lambda}_T) + \bar{\lambda}_T(J_{\min} - \mathbb{E}_{\pi^*_{\bar{\lambda}_T}}[r_2^*])$. With strong duality this yields that the primal sub-optimality is upper bounded by the sum of dual sub-optimality and constraint violation. $\square$

The explicit finite-sample bounds in Theorem 2 demonstrate a trade-off between statistical error $O(\sqrt{d/N})$ and optimization error $O(1/\sqrt{T})$. In practice, $T$ should be chosen so that the optimization error matches the statistical error, yielding a balanced trade-off among estimation accuracy, data coverage, constraint slack, and algorithmic complexity.

## 6 Simulation

We simulate in a finite prompt-action environment where features are drawn as $\phi(x,a) \sim \mathcal{N}(0, I_d)$ and normalized to unit $L_2$-norm. Ground-truth parameters $\theta_1^*$ and $\theta_2^*$ are independently sampled from $\mathcal{N}(0, I_d)$ and normalized. We set $\theta_0 = w\theta_1^* + (1-w)\theta_2^*$ and define $\pi_0(a|x) \propto$

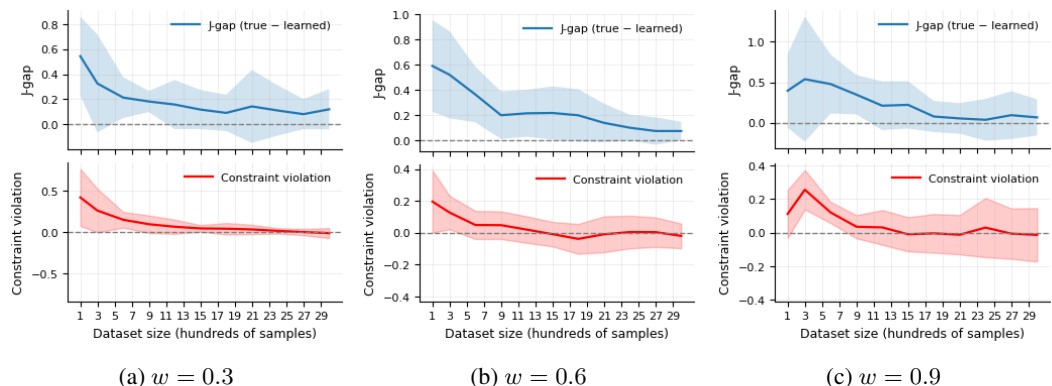

(a) $w = 0.3$      (b) $w = 0.6$      (c) $w = 0.9$

Figure 1: Performance vs.dataset size ($N$) with $T = 1000$ over three settings of $w$. **Top:** Primal objective sub-optimality. **Bottom:** Constraint violation.

$\exp(\frac{1}{\eta_0} \langle \theta_0, \phi(x, a) \rangle)$. Here, $w$ controls behavioral bias and $\eta_0$ controls coverage. We use 5 seeds, and over each random seed we generate a dataset $\mathcal{D}_{N_{\max}}$ with $|\mathcal{D}_{N_{\max}}| = 3,000$ samples by repeating: (1) sample $x$ uniformly from $\mathcal{X}$, (2) sample $a, a' \overset{i.i.d}{\sim} \pi_0(\cdot|x)$, and (3) draw Bradley-Terry preferences $y_1, y_2$ using $\theta_1^*, \theta_2^*$.

For each random seed, we evaluate convergence by measuring performance on the first $N$ samples of the dataset, increasing $N$ from 0 to $N_{\max} = 3000$ in increments of 300. In our simulations, we set $|\mathcal{X}| = 100$ and $|\mathcal{A}| = 10$ to reduce computational overhead. For each oracle $k \in \{1, 2\}$, we estimate $\widehat{\theta}_k$ using a regularized Bradley–Terry MLE (for stability at small $N$), applied to the pairwise feature differences $\phi(x, a) - \phi(x, a')$, optimized with L-BFGS. We minimize the empirical dual using projected gradient descent as in Algorithm 1 with step size $\alpha = \frac{\eta}{B^2}$ where $\eta = .05$ and $B = \max_{x,a} |\langle \widehat{\theta}_2, \phi(x, a) \rangle|$.

To generate an active yet feasible constraint level, we calibrate $J_{\min}$ once per configuration using the ground-truth reward and an expanded dataset of 10,000 samples to ensure stability: $J_{\min} = E_0 + \text{frac} \cdot (E_{hi} - E_0)$, where $E_0$ is the constraint expectation at $\lambda = 0$ and $E_{hi}$ is the expectation at $\lambda_{hi} = 5$. Each figure reports averages over random seeds, using the same per-seed dataset prefixes across parameter configurations. When varying $w$, we regenerate $\pi_0$ and the per-seed datasets, and recalibrate $J_{\min}$ accordingly. For comparison, we approximate $\lambda^*$ via gradient descent using ground-truth rewards with increased iterations for accuracy, and then recover the corresponding optimal Gibbs policy as in our analysis.

Our simulations confirm the theory: Figure 1 shows convergence across three values of $w$, with both primal suboptimality and constraint violation decreasing as $N$ increases. Shaded regions indicate confidence intervals over 5 random seeds, illustrating the consistency of our approach. As $w$ increases from 0.3 (which biases the generating policy toward oracle 1) to $w = 0.9$ (which biases toward oracle 2), we observe a higher initial constraint violation. In each setting, however, we observe convergence to near-zero violation and suboptimality.

## 7    CONCLUSION AND FUTURE WORK

We studied constrained RLHF in the offline setting with two reward oracles, proposed a dual-only method that reduces learning to a one-dimensional convex program with a closed-form Gibbs policy, and derived finite-sample guarantees that separate optimization error (tuned by $T$) from a statistical floor (governed by coverage and $N$). Simulations on synthetic prompt–action environments corroborate the theory, showing simultaneous decay of primal suboptimality and constraint violation. As future work, we aim to extend our results to richer preference models, exploit cross-oracle dependence via joint estimation, address multiple constraints and distribution shift through robust formulations, and develop online constrained RLHF.

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
