# OpenReview forum: "Constrained Preference RLHF"
_ICLR.cc/2026/Conference — Submitted to ICLR 2026_

### Official Review · Reviewer_Afcc · 2025-10-23

**Soundness:** 3
**Presentation:** 2
**Contribution:** 2
**Rating:** 4
**Confidence:** 3

**Summary:**

This paper studies offline constrained RL from human feedback with multiple preference oracles, motivated by balancing performance with safety/fairness.
The authors propose a dual-only procedure that jointly updates the policy and the Lagrange multiplier, and derive non-asymptotic guarantees.
Their analysis shows that dataset coverage is the key driver of both the optimality gap and the constraint violation of the learned policy.

**Strengths:**

The paper’s main contribution is the first finite-sample guarantee for offline constrained RLHF with multiple reward oracles. The problem formulation is interesting. Nevertheless, the paper in its current form requires additional revision and polishing before it can be considered for acceptance.

**Weaknesses:**

The main weakness of this paper lies in its lack of completeness. Specifically, it does not include thorough comparisons with prior work, presents too few experimental results (e.g., lacks comparisons with other algorithms and ablation studies over key parameters such as $T$, $B$, and $|\mathcal{A}|$), and provides insufficiently detailed derivations in the proofs.

- The motivation and real-world use case are unclear. Please clarify a concrete scenario in which your setting is warranted. In RLHF, even with multiple groups, it is common to train a single reward model; training separate group-specific rewards appears naive without stronger justification.

- The work presents a hard-constraint formulation, but Theorem 2 and the experiments appear to allow some violations. It may be clearer to reframe the problem as controlling/penalizing violations instead of enforcing a strict hard constraint.

- In Theorem 2, the upper bound on the constraint violation does not vanish even when $J_\min = 0$. This makes the theoretical guarantee rather weak, as one would expect zero violation in this case.

- The upper bounds in Theorem 2 depend on the projection radius parameter $R$. When  $R=0$, the bounds appear to become very small. Is this interpretation correct? Intuitively, it seems that we should incur some penalty when $R$ is small.

- The claim that the statistical error is $O(\sqrt{d/N})$ is overstated. Because $\beta_N \approx \sqrt{\lambda_{red}B^2} $ for large $N$, the bound need not decay as $N^{-1/2}$.

- The claim that the initial constraint violation increases with larger $w$ is incorrect. In the experiments, when $w = 0.3$, the violation is approximately $0.4$, whereas for $w = 0.9$, it is around $0.2$.

**Questions:**

- What is the motivation for using Gibbs policy updates instead of a primal update? Please clarify why a primal step is not necessary in your setting.

- In Figure 1, the “violation” appears to go below zero. How is this possible?

---

> ### Author Response · Authors · 2025-11-19
>
> We thank the reviewer for the careful and constructive evaluation of our work, as well as for recognizing the significance of establishing the first finite-sample guarantees for offline constrained RLHF with multiple preference oracles. We appreciate the clear articulation of the areas in which the paper requires further development. The feedback is valuable, and we are revising the paper to address their concerns.
>
> ---
>
> ## Response to Weaknesses
>
> - The reviewer is correct that standard RLHF typically trains a single aggregated reward model. However, our setting is motivated by scenarios where explicit constraints are required. For example, consider two populations with significantly different weights in the aggregated reward. A policy optimized solely for the dominant population may perform poorly for the minority group. A second motivation comes from the Safe-RLHF literature, where the goal is to optimize a reward model while guaranteeing certifiable performance on safety-critical or harmful content. We will expand on these motivations in the revision and include references that have examined related issues through numerical simulations.
>
> - Our algorithm converges to a policy that satisfies the hard constraints. While intermediate iterates are not guaranteed to satisfy the constraints exactly, we provide finite-time bounds and convergence guarantees that depend on the size of the dataset. As the dataset size grows, and under appropriate parameter choices, the learned policy converges to the constraint-satisfying optimum. Our treatment of "hard constraints" matches the conventional interpretation used across the CMDP literature, where temporary violations during optimization are allowed but the limiting policy is guaranteed to satisfy the constraints. We also emphasize that temporary constraint violations are primarily a concern in online settings. In our offline setting, the algorithm operates on a fixed dataset, and what ultimately matters is the performance of the final policy it outputs. Temporary violations during intermediate optimization steps therefore do not pose practical issues, as long as the final policy satisfies the constraints.
>
> - The reviewer notes that in Theorem 2 the constraint violation does not vanish at $J\_{\min} = 0$ and claims this makes the theoretical analysis weak. In the case that $J\_{\min} = 0$, we have the unconstrained problem addressed in [1]. Our analysis is for the general constrained problem. As such if we trivially set $J\_{\min} = 0$ then our algorithm converges to the unconstrained optimal policy as $N,T \to \infty$, and any policy has zero constraint violation.
>
> - Similarly if $R$ is very small, by our analysis we have $\lambda^\*$ is very small with high probability. Note $R$ is chosen from Theorem 1 to be a high probability upper bound for $\lambda^\*$. If $R = 0$ we again have with high probability that $\lambda^\* = 0$ so any policy achieves zero constraint violation. It would actually be ideal that $R$ is very small, as this would imply that the constraint is very easy to satisfy. The key is that $R$ is not a tuneable hyperparameter. It is a high probability upper bound for $\lambda^\*$ which we can determine from problem parameters.
>
> - The reviewer claims that statistical error of $O(\sqrt{d/N})$ is overstated because $\beta\_N \approx \sqrt{\lambda\_{\mathrm{reg}}B^2}$. We would like to point out that the common choice of regularization parameter is $\lambda\_{\mathrm{reg}} \propto N^{-\frac{1}{2}}$. We will make it clear in the paper, addressing their concern.
>
> - We appreciate the reviewer correctly pointing out the typo in here: the claim of increasing constraint violation with increasing $w$ is incorrect. We will fix the typo.

---

> > ### Author Response · Authors · 2025-11-19
> >
> > ## Response to Questions
> >
> > - The KL regularized primal admits an optimal policy known to be of closed Gibbs form. For any $\lambda$ the optimal policy is given by
> >
> > $$\pi(a \mid x) = \frac{\pi\_0(a \mid x) \exp\left(\frac{1}{\eta}\langle \theta^\*\_1 + \lambda \theta^\*\_2, \phi(x,a) \rangle \right)}{Z\_\lambda(x)}$$
> >
> > As such, instead of a primal step, we can directly compute the optimal policy with respect to each dual step. We can view our algorithm as after each dual step implicitly taking the optimal primal step by wielding the known Gibbs structure.
> >
> > - In Figure 1, the constraint violation dips below zero because, in our simulations, we do not clip negative values to zero. When a policy satisfies a constraint with slack, this appears as a "negative violation". This convention is standard in the CMDP literature and provides a clearer depiction of how much slack a policy achieves rather than truncating the metric at zero.
> >
> > ---
> >
> > We kindly ask the reviewer to consider updating their score in light of our responses. We believe we have addressed the key concerns raised, clarifying motivation, explaining the constraint interpretation, and outlining concrete revisions to improve completeness. We are also conducting additional experiments on realistic datasets to further strengthen the empirical section. We believe the theoretical derivations are complete and that we have only omitted standard textbook steps. That said, we would be happy to expand any parts of the analysis the reviewer finds insufficiently detailed and include additional derivations in the appendix. If there are remaining issues the reviewer feels are not fully resolved, we would greatly appreciate clarification so that we can continue to improve the paper.
> >
> > ---
> >
> > [1] Wei Xiong et al. "Iterative Preference Learning from Human Feedback: Bridging Theory and Practice for RLHF under KL-constraint." ICML 2024.

---

### Official Review · Reviewer_cmqF · 2025-10-31

**Soundness:** 3
**Presentation:** 3
**Contribution:** 2
**Rating:** 4
**Confidence:** 3

**Summary:**

This paper studies offline constrained reinforcement learning from human feedback (RLHF) with multiple preference oracles. The goal is to maximize a target group’s utility while guaranteeing a minimum welfare level for a protected group. The authors formulate a KL-regularized constrained optimization problem with a dual structure and derive a closed-form Gibbs (Boltzmann) form for the optimal policy, which reduces learning to a one-dimensional convex dual problem. They propose a dual-only projected-gradient method and provide theoretical guarantees for high-probability constraint satisfaction and a finite-sample optimality gap. Numerical experiments are presented to support the theory.

**Strengths:**

1.The paper introduces a novel formulation named constrained RLHF that combines ideas from constrained RL and RLHF in a coherent way.

2.The writing is clear and the notation/definitions are easy to follow.

3.The theoretical results are strong. The assumptions (linear rewards, Slater’s condition, bounded features) are standard and well-justified, and the paper provides a non-asymptotic finite-sample bound.

**Weaknesses:**

1.The experimental evaluation is limited. Only Gaussian-feature simulations are reported; there are no tests on realistic human-feedback datasets or environments, and no comparisons against existing methods.

2.The algorithmic novelty is modest. The “dual-only” approach is essentially projected gradient descent applied to a known dual structure.

3.The linear-reward and full-support assumptions may limit applicability to large-scale RLHF scenarios where preference models are neural and coverage is incomplete.

**Questions:**

1.How does your dual analysis differ from standard CMDP treatments? Does the KL term specifically enable the reduction to a 1-D dual variable, or would a similar reduction hold without KL regularization?

2.There are related works on “Safe RLHF” and similar formulations. Could you compare problem setups and guarantees (assumptions, types of constraints, and sample-complexity/regret guarantees) more explicitly?

3.Is it feasible to add comparisons with CMDP and RLHF baselines on more complex benchmarks (e.g., preference datasets or simulated environments with partial coverage)? If space is tight, pointers to an appendix with additional experiments would help.


While the formulation is new, it appears to directly combine RLHF with CMDP machinery, and the analysis seems to lean heavily on existing results. Could you clarify the specific theoretical innovations beyond recombining known components? In addition, the experimental section is quite limited.

---

> ### Author Response · Authors · 2025-11-19
>
> We thank the reviewer for the detailed feedback and for highlighting both the strengths and shortcomings of our paper. We acknowledge the concern regarding the limited scope of our simulations and the lack of a thorough comparison with the Safe RLHF literature. We are currently conducting more extensive experiments using realistic datasets and will expand our discussion to situate our work more clearly within the existing literature. We believe that this round of review has helped us position our contribution more effectively.
>
> ---
>
> ## Response to Weaknesses
>
> 1. We view our work as establishing foundations that enable future implementations. We will add discussion of these limitations. For comparison to existing methods, we include the following table:
>
>    ### Comparison Table
>
>    | **Method** | **Objective** | **Setting** | **Guarantees** | **Constraints** |
>    |------------|---------------|-------------|----------------|-----------------|
>    | [11] | $\max\_\pi \min\_k \mathbb{E}\_\pi[r\_k]$ | Online | None | None (max-min) |
>    | [12] | $\max\_\pi \min\_{P \in \mathcal{U}} \mathbb{E}\_P[r]$ | Offline | None | None (robustness) |
>    | [13] | N/A (axiomatic) | Theoretical | Axiomatic properties | N/A |
>    | **Ours** | $\max \mathbb{E}[r\_1]$ s.t. $\mathbb{E}[r\_k] \geq J\_{k,\min}$ | **Offline** | **Finite-sample** | **Hard (certified)** |
>
> 2. We do agree that the dual only approach is essentially projected gradient descent, we see this as a strength rather than a weakness. That constrained RLHF with KL regularization admits a closed-form Gibbs policy via a low-dimensional dual is not obvious, and it is not at all a weakness that our reduction leads to a solution as elegant as projected gradient descent. Our contribution is showing that this combination admits tractable finite-sample analysis via the dual structure. This is analogous to how [10] showed unconstrained RLHF admits finite sample analysis. Both are combinations of known ideas, but the analysis is non-trivial.
>
> 3. We emphasize that our assumptions are identical to those in the foundational offline RLHF literature and are necessary for tractable finite-sample analysis. The full coverage assumption is standard in offline RLHF [7,9,10], as well as is modeling the reward as linear [5,6,8,10]. Works such as [2] demonstrate that linear reward models achieve strong performance in practice. The existence of a strictly feasible policy is necessary for strong duality in constrained optimization, and is standard in constrained RL [1,3,4]. Moreover we provide Corollary 1 which checks Slater's condition with high probability from the offline dataset. We view our work as establishing what is provably achievable under standard assumptions. Necessity of these assumptions is important future work but beyond our scope.

---

> > ### Author Response · Authors · 2025-11-19
> >
> > ## Response to Questions
> >
> > 1. We emphasize that our framework is not limited to KL regularization. The dual reduction with closed-form policy extends to any $f$-divergence where the regularized objective admits an analytic solution. Specifically:
> >
> >    - $\alpha$-divergence: $\pi^\*\_\lambda(a|x) \propto \pi\_0(a|x) \left[1 + \frac{(\alpha-1)(r\_1^\* + \lambda r\_2^\*))}{\eta}\right]\_+^{\frac{1}{\alpha-1}}$
> >    - $\chi^2$-divergence: $\pi^\*\_\lambda(a|x) \propto \pi\_0(a|x) \left[1 + \frac{r\_1^\* + \lambda r\_2^\*}{2\eta}\right]\_+$
> >
> >    $KL$ divergence (corresponding to $\alpha \to 1$) is the most common choice in RLHF, which is why we focus on it. In contrast, standard CMDP methods without divergence regularization do not admit such closed forms, requiring iterative primal-dual methods. The divergence regularization is what enables the dual reduction.
> >
> > 2. We will include the table comparing our work with the most relevant and recent Safe RLHF papers, and we will also add a discussion that clearly highlights our contributions relative to theirs.
> >
> > 3. We are working on simulations using realistic datasets to assess whether our theoretical guarantees translate into strong empirical performance. We agree that such experiments are essential for demonstrating the practical effectiveness of our theoretical contributions.
> >
> > 4. We believe our theoretical contributions go beyond a straightforward combination of existing results, and that the insights we develop are not present in the current literature. In particular:
> >
> >    a. First formal treatment of offline constrained RLHF with multiple preference oracles and finite-sample guarantees. Safe-RLHF papers examine a related problem but provide no theoretical analysis, whereas we give a full finite-sample treatment in an explicitly constrained setting.
> >
> >    b. That KL-regularized constrained RLHF admits a dual with closed-form Gibbs policy. This is not a generic CMDP result, it exploits the specific structure of preference-based rewards and KL regularization. We extend the standard RLHF duality argument to the constrained case, which, to our knowledge, has not been established before.
> >
> >    c. Analysis of how MLE error in multiple oracles flows through the dual function to affect constraint satisfaction. Standard RLHF concentration results [10] do not address constraints. Standard CMDP results mostly assume known rewards.
> >
> >    d. Computable check for Slater's condition from finite samples.
> >
> >    We acknowledge our work builds on established techniques (Bradley-Terry MLE, Lagrangian duality, projected gradient descent). However, the combination and analysis for offline constrained preference learning are novel and non-trivial.
> >
> > ---
> >
> > We kindly ask the reviewer to consider updating their scores in light of our responses. While we acknowledge the current limitations in our experimental evaluation, we believe we have addressed all other feasible concerns raised in the review, and we are actively working on adding simulations using realistic datasets. If the reviewer still finds the work lacking in specific ways, we would greatly appreciate clarification on the remaining concerns so that we can further improve the paper.
> >
> > ---
> >
> > ## References
> >
> > [1] Eitan Altman. Constrained Markov Decision Processes. Chapman & Hall/CRC, 1999.
> >
> > [2] Paul F. Christiano et al. "Deep reinforcement learning from human preferences." NeurIPS 2017.
> >
> > [3] Dongsheng Ding et al. "Natural Policy Gradient Primal-Dual Method for Constrained Markov Decision Processes." NeurIPS 2020.
> >
> > [4] Yonathan Efroni, Shie Mannor, and Matteo Pirotta. "Exploration-exploitation in constrained mdps."
> >
> > [5] Ellen Novoseller et al. "Dueling Posterior Sampling for Preference-Based Reinforcement Learning." UAI 2020.
> >
> > [6] Aadirupa Saha, Aldo Pacchiano, and Jonathan Lee. "Dueling rl: Reinforcement learning with trajectory preferences." AISTATS 2023.
> >
> > [7] Wei Xiong et al. "Iterative Preference Learning from Human Feedback: Bridging Theory and Practice for RLHF under KL-constraint." ICML 2024.
> >
> > [8] Yichong Xu et al. "Preference-based reinforcement learning with finite-time guarantees". NeurIPS 2020.
> >
> > [9] Wenhao Zhan et al. "Provable Offline Preference-Based Reinforcement Learning." ICLR 2024.
> >
> > [10] Banghua Zhu, Jiantao Jiao, and Michael I. Jordan. "Principled Reinforcement Learning with Human Feedback from Pairwise or $K$-wise Comparisons".
> >
> > [11] Chakraborty, Souradip, et al. "MaxMin-RLHF: alignment with diverse human preferences." ICML 2024.
> >
> > [12] Ramesh, Shyam Sundhar, et al. "Group robust preference optimization in reward-free rlhf." NeurIPS 2024.
> >
> > [13] Ge, Luise, et al. "Axioms for ai alignment from human feedback." NeurIPS 2024.

---

### Official Review · Reviewer_u4Ty · 2025-11-01

**Soundness:** 3
**Presentation:** 2
**Contribution:** 1
**Rating:** 2
**Confidence:** 4

**Summary:**

This paper studies the problem of constrained RLHF. The model of RLHF considered is the standard Bradley-Terry model, and the reward function $r\_1$ in linear in some feature vector $\theta$. The paper extends standard RLHF to the setting where there is a second reward function $r\_2$, and the policy is constrained to obtain some minimum reward level with respect to $r\_2$, while maximizing the standard rewards $r\_1$. Both these reward functions must be initially learned from preference data, after which the constrained optimization must be solved. The paper proposes an algorithm that leverages closed-form optimization of the primal policy in order to iteratively solve the dual problem.

**Strengths:**

Fairness in RLHF is an important topic, and the algorithm that this paper develops seems sound for the case of a single additional reward constraint.

**Weaknesses:**

This paper studies a very limited model of constrained RLHF, and simply combines known results regarding reward estimation with standard constrained RL methods. In more detail, this paper studies a single additional reward constraint that is supposed to represent the utility of some protected population. However, a large body of work in fair RLHF already studies the more complex and realistic setting of multiple protected populations. For a few examples of such research see [1,2,3]. The setting of multiple protected populations is much more interesting both because there clearly will be multiple such population in practice, and because it requires more nuanced algorithms that have to make a choice about how to satisfy heterogeneous, possibly competing constraints. This submission does not cite any of this highly-relevant work on fairness in RLHF, and does not really compare favorably to the current body of work in the field, as described above.

To summarize, the model studied is quite limited compared to already existing theoretical work on fairness in RLHF, and the main results are straightforward applications of known concentration bounds, and constrained RL algorithms.


[1] Chakraborty, Souradip, et al. "MaxMin-RLHF: alignment with diverse human preferences." ICML 2024.

[2] Ramesh, Shyam Sundhar, et al. "Group robust preference optimization in reward-free rlhf." NeurIPS 2024.

[3] Ge, Luise, et al. "Axioms for ai alignment from human feedback." NeurIPS 2024.

**Questions:**

1. In what RLHF setting would there only be a single constraint corresponding to one protected population?

2. Is there any sense in which your algorithm is not a special case of existing multi-group RLHF methods?

---

> ### Author Response · Authors · 2025-11-19
>
> We thank the reviewer for pointing out the papers. The literature moves so quickly, and missing some important references is inevitable. We will include these references together with a table of comparisons to highlight our contribution vs theirs contribution. We have provided the table as a response as well.
>
> ---
>
> ## Extension to $m$ oracles
>
> Our work easily extends to the setting of $m$ constraints. The Lagrangian becomes
>
> $$\mathcal{L}(\pi,\boldsymbol{\lambda}) = \mathbb{E}\_\pi \left[r^\*\_1 + \sum\_{k=2}^{m+1} \lambda\_k r^\*\_k\right] - \eta D\_{\mathrm{KL}}(\pi||\pi\_0) - \sum\_k \lambda\_k J\_{k,\min}$$
>
> The optimal policy retains its closed form Gibbs structure, and all theoretical results extend with $O(m)$ dependence in error bounds. Algorithm 1 becomes projected gradient descent in $\mathbb{R}\_+^m$ with iteration complexity reflecting this. All these extensions will readily follow, and we will add a section addressing this extension to general case.
>
> ---
>
> ## Addressing the other paper
>
> We thank the reviewer for bringing these important references to our attention. We acknowledge we should have cited these recent works on fairness in RLHF and will add comprehensive discussion in revision. However, we respectfully disagree that our work is "simply combining known results" or that existing multi-group methods dominate our approach. Below we clarify the distinctions.
>
> **Reference [1]**: Maximizes $\min\_{k} \mathbb{E}\_{\pi}[r\_k]$ across $K$ groups to ensure fairness. This is an unconstrained max-min objective, not constrained optimization. Their Algorithm 1 performs PPO updates with a dynamically reweighted reward $\sum\_k w\_k r\_k$ where weights adjust to prioritize poorly-performing groups.
>
> - They optimize worst-case welfare, but provide no guarantees that any group achieves a minimum threshold
> - Empirical evaluation only
> - Max-min welfare $\neq$ maximize primary objective subject to minimum welfare constraints
>
> **Reference [2]**: Optimizes under worst-case distribution shift across groups via distributionally robust optimization (DRO). Their objective is $\max\_\pi \min\_{P \in \mathcal{U}} \mathbb{E}\_{P}[r\_\pi]$ where $\mathcal{U}$ is an uncertainty set.
>
> - Addresses uncertainty in group distributions, not explicit welfare constraints
> - DRO minimizes worst-case regret but doesn't enforce $\mathbb{E}\_{\pi}[r\_k] \geq J\_{k,\min}$
>
> **Reference [3]**: Proposes axiomatic foundations for aggregating preferences (e.g., Pareto efficiency, anonymity). Provides theoretical characterizations but no algorithms or finite-sample analysis.
>
> ### Comparison Table
>
> | **Method** | **Objective** | **Setting** | **Guarantees** | **Constraints** |
> |------------|---------------|-------------|----------------|-----------------|
> | [1] | $\max\_\pi \min\_k \mathbb{E}\_\pi[r\_k]$ | Online | None | None (max-min) |
> | [2] | $\max\_\pi \min\_{P \in \mathcal{U}} \mathbb{E}\_P[r]$ | Offline | None | None (robustness) |
> | [3] | N/A (axiomatic) | Theoretical | Axiomatic properties | N/A |
> | **Ours** | $\max \mathbb{E}[r\_1]$ s.t. $\mathbb{E}[r\_k] \geq J\_{k,\min}$ | **Offline** | **Finite-sample** | **Hard (certified)** |
>
> ---
>
> ## Our contribution
>
> Our work addresses constrained offline RLHF with provable constraint satisfaction, which none of the cited works provide. We provide finite-sample bounds on constraint violation. MaxMin-RLHF and Group Robust PO provide no such guarantees. They optimize objectives but cannot certify that group $k$ achieves welfare $\geq J\_{k,\min}$.
>
> Our work is positioned complementary to the referenced literature: they address fairness objectives via empirical simulations while we address certifiable welfare constraints from a theoretical standpoint. We will incorporate these sources into our related-work section. We will also include the table above, which helps position our contribution more clearly. We appreciate the reviewer's guidance in situating our work more accurately within the broader Safe RLHF literature.
>
> We would like to ask the reviewer to update their scores in light of responses. We believe we have addressed all of their concerns and we are currently working on adding simulations with realistic dataset. If the reviewer feels like we are still short on our results, we would like to know their main concerns so we can improve on our paper.

---

### Official Review · Reviewer_AtsF · 2025-11-02

**Soundness:** 2
**Presentation:** 2
**Contribution:** 2
**Rating:** 2
**Confidence:** 5

**Summary:**

This paper presents a linear, convex constrained optimization framework for reinforcement learning from human feedback (RLHF) using binary preference data. The authors study offline constrained RLHF where two independent preference oracles—a target population and a protected group—provide pairwise comparison feedback. The goal is to maximize the expected utility of the target oracle while ensuring the protected group’s welfare stays above a specified threshold, formulated as a convex problem with KL regularization and a single expectation constraint.

Each reward function is linear in features and estimated from binary pairwise preferences through a Bradley–Terry logistic model. The resulting problem admits a one-dimensional strongly convex dual formulation in the Lagrange multiplier. The authors propose a dual-only projected gradient descent algorithm that leverages a closed-form Gibbs policy and derive finite-sample guarantees separating statistical estimation error $O(\sqrt{d/N})$ and optimization error $O(1/\sqrt{T})$.

Theoretical results show high-probability bounds on suboptimality and constraint violation, ensuring the learned policy is nearly optimal and feasible. Simulations in a synthetic linear feature environment support the analysis, showing that both primal suboptimality and constraint violation decay with data size. The work offers a formal theoretical treatment of constrained RLHF integrating preference-based reward estimation and constraint satisfaction, though it remains limited to a linear and convex setting rather than large-scale empirical validation.

**Strengths:**

1. **KL-Regularized Dual Formulation of Constrained RLHF**

    The paper establishes a clear convex dual structure for KL-regularized constrained RLHF.

    By integrating the objective and constraint into a unified Lagrangian and proving strong duality under Slater’s condition, the authors reduce the primal problem to a one-dimensional convex dual optimization.

    This formulation enables a provably convergent dual-only projected gradient descent algorithm and provides a more principled convex foundation for constrained RLHF compared to prior heuristic approaches.

2. **Propagation Analysis of Reward Estimation Uncertainty**

    Lemma 2–3 rigorously quantify how reward estimation errors from Bradley–Terry MLE propagate into the dual function and its gradient.

    The authors derive explicit high-probability bounds on $| \hat{g}(\lambda)-g(\lambda) |$ and $| \hat{g}'(\lambda)-g'(\lambda) |$, linking statistical uncertainty to constraint satisfaction and dual stability.

    This represents a novel theoretical insight that formalizes the coupling between reward estimation accuracy and feasibility in constrained RLHF.

3. **Finite-Sample Guarantees for Optimality and Feasibility**

    Theorem 2 provides a unified finite-sample analysis that decomposes the total error into statistical and optimization components:

    $O(\sqrt{d/N})$ for reward estimation and $O(1/\sqrt{T})$ for dual optimization.

    The result delivers explicit high-probability guarantees for dual suboptimality, constraint violation, and primal optimality gap.

    This highlights the fundamental trade-off among data coverage, iteration budget, and constraint slack in constrained RLHF.


---

In summary, the theoretical development is distinctive for (1) framing KL-regularized constrained RLHF as a convex dual problem, (2) explicitly characterizing uncertainty propagation from reward estimation to dual errors, and (3) deriving the first finite-sample optimality and feasibility bounds that quantify key bottlenecks in constrained RLHF.

**Weaknesses:**

- While the paper claims to present “the first formal treatment of constrained RLHF with multiple reward oracles,” this claim seems overstated. In practice, the formulation does not extend to multiple independent oracles but instead focuses on a two-oracle structure (one target and one constraint oracle). This setup is conceptually similar to that of *Safe RLHF: Safe Reinforcement Learning from Human Feedback* (ICLR 2024), which also maximizes a reward signal under a secondary constraint derived from human feedback. The contribution of this work therefore lies primarily in its convex-theoretic formalization and finite-sample guarantees, rather than in introducing a genuinely new multi-oracle framework.
- The theoretical framework relies on highly restrictive assumptions—linear reward models, full policy coverage, and the existence of a strictly feasible policy under Slater’s condition. While these assumptions make the analysis elegant, they significantly limit the generality and applicability of the results to realistic RLHF settings, such as large-scale language models with nonlinear reward functions or partial coverage data.
- Although the paper offers a clean convex and dual analysis, its conceptual novelty remains limited. The theoretical results formalize expected convex behaviors (e.g., strong duality, Lipschitz continuity, finite-sample convergence) but do not reveal new mechanisms or interactions unique to constrained RLHF. As such, the work feels more like a rigorous theoretical consolidation of existing ideas than a conceptual leap forward.
- The paper does not deeply engage with the practical bottlenecks of applying constrained RLHF, such as instability in dual variable updates, reward–constraint coupling, or data imbalance across preference sources. These issues are mentioned but not theoretically explored, leaving a gap between the presented analysis and the empirical challenges of real-world constrained RLHF.

**Questions:**

- The paper refers to “multiple reward oracles” as a defining aspect of its contribution, but the formulation effectively uses only two oracles—one for the target population and one for the protected group. Could the authors elaborate on how the proposed framework would generalize to *more than two* oracles? For example, if several stakeholder groups with distinct preference distributions were involved, would the dual formulation still remain tractable, or would multiple dual variables and constraints be required? A clarification or theoretical sketch of how the method scales to true multi-oracle settings would greatly strengthen the paper’s claim.
- Given that the current formulation assumes linear reward models and full coverage, it would be helpful to understand how the algorithm behaves when these assumptions are relaxed. Have the authors considered testing a relaxed version of the algorithm—e.g., with approximate coverage or nonlinear (e.g., neural) reward estimators—in more practical RLHF settings such as text-generation or summarization tasks? Even small-scale experiments on realistic datasets could offer valuable insights into whether the theoretical guarantees translate into meaningful safety or fairness effects in practice.
- Relatedly, could the authors discuss the computational or stability challenges that might arise when extending the dual update or Gibbs policy to more complex preference models or partially observed constraints? It would be interesting to know whether the proposed approach can maintain convergence or constraint satisfaction guarantees under such conditions.

---

> ### Author Response · Authors · 2025-11-19
>
> We thank the reviewer for recognizing the strengths of our paper, as well as for their suggested improvements and the missing references. Their feedback helps us refine our work and better position the paper within the existing literature.
>
> ---
>
> ## Response to weaknesses
>
> - As we have pointed out in our paper, their work "can be formalized as a constrained RLHF problem involving two distinct reward oracles". However, the cited work on Safe RLHF provides no performance guarantees, and instead relies purely on empirical results in the online setting. In contrast, our work provides the first rigorous analysis of constrained RLHF particularly in the offline setting as the reviewer correctly pointed out as well: (1) Finite sample bounds on primal sub-optimality and constraint violation (Theorem 2) (2) Statistical error propagation from MLE estimates through dual function (Lemma 2) (3) Data dependent bounds on optimal dual variable (Theorem 1) and (4) Computable slack from finite samples (Corollary 1).
>
>   Our work is the first to quantify exactly the relation between estimation error and convergence error (dataset size vs sample complexity) which can provide a guidance for the choice of hyperparameters. The distinction parallels RLHF [6] vs. RLHF with regret bounds [11]. Safe RLHF deserves credit for identifying the problem and demonstrating empirical success, while our work provides the first provably correct offline solution with finite sample guarantees. We will elaborate on this distinction in our section on Related Work.
>
> - We emphasize that our assumptions are identical to those in the foundational offline RLHF literature and are necessary for tractable finite-sample analysis. The full coverage assumption is standard in offline RLHF [8,10,11], as well as is modeling the reward as linear [5,7,9,11]. Works such as [2] demonstrate that linear reward models achieve strong performance in practice. The existence of a strictly feasible policy is necessary for strong duality in constrained optimization, and is standard in constrained RL [1,3,4].
>
> - We respectfully note that several aspects of our analysis are specific to constrained RLHF, not merely applications of convex optimization. That KL-regularized constrained RLHF admits a closed form Gibbs structure is a consequence of linear rewards, Bradley-Terry preferences, and Lagrangian duality; not a generic property of constrained optimization. Our analysis reveals that poor reference policy coverage creates infeasibility, not just slower convergence. Theorem 1 shows $\lambda^\*$ scaled with $\rho^{-1}$ where the Slater slack $\rho$ degrades with $\lambda\_{\min}(\Sigma\_{N,\mathrm{reg}})$. In unconstrained RLHF, coverage affects only sample complexity, while here it determines whether tight constraints are achievable at all. The coupling is specific to offline constrained preference learning. We note that the reviewer does not specify what might constitute a conceptual leap in this context.
>
> - We appreciate the reviewer highlighting practical considerations. Our analysis directly addresses dual variable instability via Lemma 1 and Proposition 1. These properties guarantee that projected gradient descent converges at rate $O(1/\sqrt{T})$ with step size $\eta/B^2$. The strong convexity parameter is computable from data and determines convergence speed. Our analysis in addition also naturally handles imbalanced data. Since $\theta\_1^\*$ and $\theta\_2^\*$ are estimated via separate MLEs, the two oracles can provide a different number of comparisons. Nevertheless, there are many open directions remain to pursue as future work, and addressing all challenges in the first theoretical paper of its kind is simply not feasible.

---

> > ### Author Response · Authors · 2025-11-19
> >
> > ## Response to questions
> >
> > - Our work easily extends to the setting of $m$ constraints. The Lagrangian becomes
> >
> > $$\mathcal{L}(\pi,\boldsymbol{\lambda}) = \mathbb{E}\_\pi \left[r^\*\_1 + \sum\_{k=2}^{m+1} \lambda\_k r^\*\_k\right] - \eta D\_{\mathrm{KL}}(\pi||\pi\_0) - \sum\_k \lambda\_k J\_{k,\min}$$
> >
> > The optimal policy retains its closed form Gibbs structure, and all theoretical results extend with $O(m)$ dependence in error bounds. Algorithm 1 becomes projected gradient descent in $\mathbb{R}\_+^m$ with iteration complexity reflecting this. All these extensions will readily follow, and we will add a section addressing this extension to general case.
> >
> > - The reviewer is correct that our linear reward assumption imposes coupling through the shared feature space. This is a fundamental modeling choice in RLHF literature [8,11]. Relaxing to non-linear rewards or separate feature spaces for different oracles would require neural function approximation, which remains an open problem even for unconstrained offline RLHF with finite sample guarantees. We agree this is an important direction for future work. For now, we are working on adding simulations on realistic dataset and see whether our theoretical guarantees translate to empirical performance or not.
> >
> > - We would like to emphasize that the primary motivation of our work is theoretical, and our main goal is to derive performance bounds. As we have pointed out, these are interesting future directions that are somewhat orthogonal to the current work. This is the first paper that studies this problem from a theoretical standpoint and there are many different directions remain open to pursue as future work.
> >
> > We would like to ask the reviewer to update their scores in light of responses. We believe we have addressed most of their concerns, that deem feasible, and we are currently working on adding simulations with realistic dataset. If the reviewer feels like we are still short on our results, we would like to know their main concerns so we can improve on our paper.
> >
> > ---
> >
> > ## References
> >
> > [1] Eitan Altman. Constrained Markov Decision Processes. Chapman & Hall/CRC, 1999.
> >
> > [2] Paul F. Christiano et al. "Deep reinforcement learning from human preferences." NeurIPS 2017.
> >
> > [3] Dongsheng Ding et al. "Natural Policy Gradient Primal-Dual Method for Constrained Markov Decision Processes." NeurIPS 2020.
> >
> > [4] Yonathan Efroni, Shie Mannor, and Matteo Pirotta. "Exploration-exploitation in constrained mdps."
> >
> > [5] Ellen Novoseller et al. "Dueling Posterior Sampling for Preference-Based Reinforcement Learning." UAI 2020.
> >
> > [6] Long Ouyang et al. "Training Language Models to Follow Instructions with Human Feedback." NeurIPS 2022.
> >
> > [7] Aadirupa Saha, Aldo Pacchiano, and Jonathan Lee. "Dueling rl: Reinforcement learning with trajectory preferences." AISTATS 2023.
> >
> > [8] Wei Xiong et al. "Iterative Preference Learning from Human Feedback: Bridging Theory and Practice for RLHF under KL-constraint." ICML 2024.
> >
> > [9] Yichong Xu et al. "Preference-based reinforcement learning with finite-time guarantees". NeurIPS 2020.
> >
> > [10] Wenhao Zhan et al. "Provable Offline Preference-Based Reinforcement Learning." ICLR 2024.
> >
> > [11] Banghua Zhu, Michael Jordan, and Jiantao Jiao. "Principled reinforcement learning with human feedback from pairwise or k-wise comparisons." ICML 2023.

---

### Author Response · Authors · 2025-11-28
**Comments to all reviewers: Simulation Results**

We use the PKU-SafeRLHF dataset containing 73,907 training examples with dual preference labels for both safety and helpfulness. Each example consists of a prompt and two responses generated by Alpaca-7B, Alpaca2-7B, or Alpaca3-8B, with human annotations indicating which response is safer and which is more helpful. We convert each (prompt, response) pair into a 768-dimensional feature vector using Sentence-BERT (all-mpnet-base-v2) and normalize to unit $L\_2$ norm: $\|\phi(x,a)\|\_2 = 1$.

We estimate linear reward parameters $\hat{\theta}\_1, \hat{\theta}\_2 \in \mathbb{R}\^{768}$ via Bradley-Terry maximum likelihood estimation using L-BFGS-B optimization with $L\_2$ regularization coefficient $\lambda\_{\text{reg}} = 0.01$. Safety rewards use $r\_1(x,a) = \hat{\theta}\_1\^\top \phi(x,a)$ estimated from safety preference labels, while helpfulness rewards use $r\_2(x,a) = \hat{\theta}\_2\^\top \phi(x,a)$ estimated from helpfulness preference labels. We compute the reference policy $\pi\_0$ using Alpaca-7B. For each response, we compute the log-likelihood via autoregressive decomposition:
$$
\log \mathbb{P}(\text{response} \mid \text{prompt}) = \sum\_{t=1}\^T \log \mathbb{P}(\text{token}\_t \mid \text{token}\_{<t}, \text{prompt})
$$
We apply length normalization (dividing by $T$) to obtain comparable log-probabilities $\ell\_0$ and $\ell\_1$ for response 0 and response 1, respectively. The reference policy is then:
$$
\pi\_0(\text{response}\_0 \mid x) = \frac{\exp(\ell\_0)}{\exp(\ell\_0) + \exp(\ell\_1)}
$$
computed using the log-sum-exp trick for numerical stability. This normalization is necessary because raw probabilities underflow to zero for multi-token sequences. The resulting $\pi\_0$ is a valid probability distribution over the binary action space $\{\text{response}\_0, \text{response}\_1\}$ for each prompt. We maximize safety (objective: $\mathbb{E}\_{\pi}[r\_1]$) subject to a minimum helpfulness constraint ($\mathbb{E}\_{\pi}[r\_2] \geq J\_{\min}$). To set $J\_{\min}$, we first compute the maximum achievable helpfulness by evaluating the greedy policy that always selects the response with higher $r\\_2$. We then set:
$$
J\\_{\min} = \mathbb{E}\\_{\pi\\_0}[r\_2] + \gamma \cdot (\mathbb{E}\_{\text{greedy}}[r\_2] - \mathbb{E}\_{\pi\_0}[r\_2])
$$
where $\gamma = 0.7$ requires achieving 70\% of the improvement gap between the reference policy and the greedy policy.

We solve the dual problem via projected gradient descent on the dual variable $\lambda \geq 0$. At each iteration $t$, we compute the Gibbs policy:
$$
\pi\_\lambda(a \mid x) = \frac{\pi\_0(a \mid x) \exp\left(\frac{r\_1(x,a) + \lambda r\_2(x,a)}{\eta}\right)}{Z(x, \lambda)}
$$
where $\eta = 0.3$ is the temperature parameter and $Z(x, \lambda)$ is the partition function. The dual gradient is $g'(\lambda\_t) = \mathbb{E}\_{\pi\_{\lambda\_t}}[r\_2] - J\_{\min}$, and we update:
$$
\lambda\_{t+1} = \text{proj}\_{[0,R]}\left(\lambda\_t - \alpha\_t g'(\lambda\_t)\right)
$$
using AdaGrad-normalized adaptive step sizes to accelerate convergence. We return the policy at the time-averaged dual variable $\bar{\lambda}\_T = \frac{1}{T}\sum\_{t=1}\^T \lambda\_t$, which provides theoretical guarantees on constraint satisfaction.

The resulting figures are available here: https://tinyurl.com/m7x3v3xp

**Finally**, we emphasize that these experiments represent preliminary simulations designed to validate the core algorithmic behavior. We are currently extending our evaluation to additional datasets and conducting more comprehensive simulations, including larger-scale models and richer forms of preference data. These extended experiments are ongoing and will be included in the final version of the paper.

---

### Author Response · Authors · 2025-12-01
**Summary of Rebuttal Process for AC**

In light of the recent developments, we have compiled a summary of the reviewers' questions and identified weaknesses, together with our detailed responses.

## Theoretical Contribution and Novelty

- **Claim that work is similar to RLHF with limited novelty:** We point out that the cited works on safe RLHF provide no theoretical guarantees, with results being purely empirical and in the online setting. We provide the first finite-sample guarantees in the offline setting.

- **Claim of straightforward application of convex optimization:** We note that the closed-form Gibbs structure is not obvious, safe RLHF misses this structure despite solving a similar problem. Our analysis pins down the relation between data coverage, estimation error and rate of convergence rate.

- **Concerns on algorithmic novelty:** The elegant approach of projected gradient descent is a strength rather than a weakness. Finite sample analysis is nontrivial, and propagation of MLE error through dual function and connecting dual optimization error to primal constraint violation requires problem specific insights.

- **Questioning specific theoretical innovations:** We give the first formal characterization of offline constrained RLHF with finite sample guarantees. We present a closed form Gibbs structure for the policy, which is not obvious in the constrained case. Our analysis extends to Multi-oracle setting.

## Comparison to Related Work

- **Concern on missing citations:** We added a comprehensive comparison table showing our work is complementary to existing works in Safe RLHF.

- **Concern that existing multi-group methods dominate our approach:** We note the differing objectives between existing works and ours, they optimize fairness/robustness while we enforce minimum thresholds with certified satisfaction. Their work cannot certify a certain performance threshold for each group k.

## Experiments and Validation

- **Concerns on limited simulation:** We added simulation using the PKU-SafeRLHF dataset showing that our algorithm performs well. We are working on extending it to other dataset.

## Assumptions and Generality

- **Concerns about restrictive assumptions:** We note that our assumptions are common in RLHF literature, necessary for tractable analysis, and provide citations for which all our assumptions are mirrored.

- **Limitation to single constraint:** We demonstrate that the assertion that our paper handles only the single constraint is incorrect, showing that multiple constraints hold trivially by Lagrangian duality.

- **Questioning the use of only KL-regularization:** We show that our work extends to any f-divergence with closed form solution. KL is however the most common in RLHF. CMDP without divergence regularization requires an iterative primal-dual approach, while divergence regularization enables a reduction to dual only algorithm.

## Technical Concerns

- **Dual variable instability:** Follows directly via results in the paper.

- **Concern that coupled feature spaces are impractical:** We note that this is a fundamental modeling choice in RLHF literature. Relaxing to neural rewards or separate features requires neural approximations, an open problem even for unconstrained offline RLHF with guarantees.

- **Concern that constraint violation doesn't vanish:** We note that constraint violation does indeed vanish as N,T → ∞ as stated in the paper.

- **Concern that constraint gap doesn't vanish with J_min = 0:** Our bound correctly identifies λ* = 0 (see the deterministic bound in Theorem 1). Consequently, the natural choice for the projection radius is R = 0, and this behavior is appropriately captured in Theorem 2. When J_min = 0 our algorithm recovers the unconstrained optimum as N,T → ∞.

- **Confusion about the role of R in finite sample bounds:** We highlight that R is not a tunable parameter, it is an upper bound on λ* from problem parameters. Small R implies constraint is easy to satisfy, while large R implies constraint is harder to satisfy. This is intuitive.

- **Claim that statistical error is overstated:** We show that when we take λ_reg = 1/√N as is common choice of regularization parameter, our statistical error is as stated in the paper.

- **Concern that figure 1 shown negative violation:** violation is not clipped at 0 in the plot. This allows us to show when the constraint holds with slack.

## Motivation and Presentation

- **Concern that the problem is not well motivated:** We include specific real world problem that are handled through constrained preference RLHF.

- **Confusion on hard vs. soft constraints:** We give the standard CMDP interpretation where intermediate violations are allowed during optimization and only the final policy must satisfy constraints with high probability. The offline setting makes intermediate violations irrelevant.

---

### Meta-Review · Area_Chair_ieR7 · 2025-12-26

**Summary:**

The authors aim to maximize the utility for a target population while ensuring a minimum welfare level for a protected group. The approach formulates a constrained optimization problem characterized by KL-regularization, presenting it as a dual problem that facilitates a closed-form Gibbs policy solution. The paper goes on to propose a dual-only projected gradient descent algorithm, offering theoretical guarantees related to constraint satisfaction and finite-sample performance. The authors analyze how factors like estimation error and data coverage influence feasibility and optimality

**Reviewer Concerns:**

Lack of Novelty: Some reviewers felt that the work does not present a significant theoretical advancement beyond existing research in constrained RLHF. Specifically, it was noted that the paper primarily combines known methods without introducing genuinely new insights.

As the AC, I have gone through the paper myself and, while novelty is subjective, I concur with the reviewers that the techniques in the paper are somewhat routine. The problem itself is interesting to study, but the convergence analyses follow from a somewhat straightforward manner via convex optimization and PGD.

Limited Experimental Validation: Reviewers highlighted the lack of extensive empirical validation, urging for comparisons against established methods and simulations on more realistic datasets rather than synthetic examples.

I have also perused the experimental section and agree with the reviewers that for a work on RLHF, the current set of experiments is not sufficiently convincing for a venue like ICLR. A larger scale experiment, perhaps with better motivation for the need for constraints, would enhance the paper.

Restrictive Assumptions: Concerns were raised regarding the reliance on linear reward models and full policy coverage. Reviewers pointed out that these assumptions limit applicability to practical RLHF scenarios, particularly those involving non-linear models or incomplete data coverage.

Motivation Clarity: Some reviewers suggested that the practical motivations for using separate reward oracles and constraints were inadequately justified. They called for more concrete examples of real-world scenarios where the presented approach would be applicable.

I have also read the introduction to understand the authors' motivation for incorporating constraints into the RLHF framework. The authors use a fairness example to justify the constraint involving $r_2$. However, it is not clear why this constraint corresponds to a **fairness**. A better motivation would be beneficial.

Clarity on Hard Constraints: There was confusion over how hard constraints were handled, with reviewers suggesting the approach could benefit from framing the problem in terms of controlling or penalizing constraint violations rather than strictly enforcing them.

I agree that these are two different ways of handling constraints. The authors' response is fine as is.

Theoretical Guarantees: Some reviewers questioned the robustness of the theoretical guarantees, especially in cases where the constraints do not vanish as expected. Concerns included the need for clearer interpretations of the bounds provided in the analysis.

The authors justified this clearly, by mentioning that $\lambda^*=0$, hence, convergence is guaranteed, even for the constraints.

Dual Variable Instability: Questions about potential instability with dual variable updates and the implications for the learned policies were raised.

This was convincingly justified.

**Reviewer Scores:**

I believe some of low scores of 2 would have probably been increased to 3 or 4. However, I am concerned that this paper is not at the level to be accepted primarily because of the following three reasons which I've mentioned above.

Theoretical Contributions: While the paper claims to provide the first formal treatment of constrained RLHF with multiple preference oracles, the analysis relies heavily on known results and does not convincingly establish novel insights. The formalization appears to be an extension of existing moderated RLHF frameworks rather than a breakthrough contribution.

Insufficient Empirical Results: The experimental evaluation is limited, showcasing only preliminary synthetic results without robust comparisons to state-of-the-art methods or realistic datasets. This inadequacy diminishes the paper's practical relevance and undermines the theoretical contributions presented.

Weak Justification for Motivation: The authors did not adequately justify their selection of a dual-only methodology or the specific need for multiple reward oracles, which undermines the foundational premise of the research.

---

### Decision · Program_Chairs · 2026-01-26

Reject